# Streptozotocin-Induced Diabetes Causes Changes in Serotonin-Positive Neurons in the Small Intestine in Pig Model

**DOI:** 10.3390/ijms23094564

**Published:** 2022-04-20

**Authors:** Michał Bulc, Katarzyna Palus, Jarosław Całka, Joanna Kosacka, Marcin Nowicki

**Affiliations:** 1Department of Clinical Physiology, Faculty of Veterinary Medicine, University of Warmia and Mazury in Olsztyn, Oczapowski Str. 13, 10-718 Olsztyn, Poland; katarzyna.palus@uwm.edu.pl (K.P.); calkaj@uwm.edu.pl (J.C.); 2Department of Visceral, Transplant, Thoracic and Vascular Surgery, University of Leipzig Medical Center, Liebigstr. 21, 04103 Leipzig, Germany; joanna.kosacka@medizin.uni-leipzig.de; 3Institute of Anatomy, University of Leipzig, Liebigstraße 13, 04103 Leipzig, Germany; marcin.nowicki@medizin.uni-leipzig.de

**Keywords:** serotonin, enteric nervous system, diabetes, pig

## Abstract

Serotonin (5-hydroxytryptamine or 5-HT) is an important neurotransmitter of the central and peripheral nervous systems, predominantly secreted in the gastrointestinal tract, especially in the gut. 5-HT is a crucial enteric signaling molecule and is well known for playing a key role in sensory-motor and secretory functions in the gut. Gastroenteropathy is one of the most clinical problems in diabetic patients with frequent episodes of hyperglycemia. Changes in 5-HT expression may mediate gastrointestinal tract disturbances seen in diabetes, such as nausea and diarrhea. Based on the double immunohistochemical staining, this study determined the variability in the population of 5-HT-positive neurons in the porcine small intestinal enteric neurons in the course of streptozotocin-induced diabetes. The results show changes in the number of 5-HT-positive neurons in the examined intestinal sections. The greatest changes were observed in the jejunum, particularly within the myenteric plexus. In the ileum, both de novo 5-HT synthesis in the inner submucosal plexus neurons and an increase in the number of neurons in the outer submucosal plexus were noted. The changes observed in the duodenum were also increasing in nature. The results of the current study confirm the previous observations concerning the involvement of 5-HT in inflammatory processes, and an increase in the number of 5-HT -positive neurons may also be a result of increased concentration of the 5-HT in the gastrointestinal tract wall and affects the motor and secretory processes, which are particularly intense in the small intestines.

## 1. Introduction 

Serotonin (5-hydroxytryptamine, 5-HT) was first isolated from the platelets in 1948 and identified as an agent causing vascular smooth muscle contraction. 5-HT is synthesized in the body by the enzymatic conversion (hydroxylation and decarboxylation) of tryptophan [1,2,3]. The synthesis of 5-HT takes place primarily in the central and peripheral nervous system neurons, chromophilic cells of the gastrointestinal tract, and mast cells [4]. In mammalian bodies, the vast majority of serotonin is found in the gastrointestinal tract, mainly in the intestinal region. Much less serotonin is found in the platelets, and only 1–2% is found in the brain [5,6]. In the gastrointestinal tract, enteroendocrine (EE) cells are scattered throughout the mucosal lining of the gut wall and synthesize and release a great number of different natural biologically active substances. Approximately half of all EE cells are responsible for the production of serotonin and are called enterochromaffin (EC) cells. EC cells are the largest source of 5-HT in the body, producing about 95% of total body 5-HT. Significantly lower amounts of 5-HT are secreted in other peripheral tissues and serotonergic neurons within the enteric neurons [6,7,8]. 5-HT synthesized in the intestinal cells is released into the blood and into the intestinal lumen. This amine exerts its biological effect after binding to the appropriate membrane receptor [9,10]. Five different 5-HT receptor types have been described in the gastrointestinal wall. They are found in the enterocytes, enteric neurons, smooth muscle cells, and immune cells [11]. The physiological role of 5-HT in the gastrointestinal tract involves a number of secretory, motor, and sensory processes. It is involved in fluid and mucin secretion within the intestines and affects the secretion of pancreatic juice [12]. Another important function is the involvement in electrolyte resorption. 5-HT also stimulates intestinal motor activity and is involved in shaping the pattern of intestinal contractile activity [13,14]. The involvement of 5-HT is not limited to the regulation of physiological processes since, due to its effects on the immune cells, it is also involved in inflammatory processes and autoimmune disorders such as inflammatory bowel disease (IBD) and coeliac disease [15]. 

The multidirectional effect of 5-HT on the gastrointestinal tract is also due to the complex innervation of this tract. Neural regulation of the gastrointestinal tract activity is carried out by exogenous neurons represented by the sympathetic and parasympathetic parts of the nervous system and by neurons that are part of the enteric neurons [16]. The enteric neurons are a highly organized structure formed by plexuses organized into ganglia. Within the intestinal region, a distinction can be drawn between the myenteric plexus and submucosal plexus, which, in large animals, is secondarily divided into the inner and outer submucosal plexus (Figure 1) [17]. The myenteric plexus is clearly visible along the entire length of the digestive tract, from the esophagus to the anal sphincter, and primarily controls the activities of the muscular layer, although some neurons supply the mucosa, blood vessels, and glands of the digestive tract and they also create connections with other types of ENS plexuses. The myenteric plexus shows significant variations depending on the species and part of the gastrointestinal tract [16,17,18]. The submucosal plexus in large mammals is divided into two types of plexuses, which are clearly visible mainly in the small and large intestines. Occasionally, they can be found in the stomach and esophagus. In turn, they are absent in the esophagus and forestomachs of ruminants. Similar to the myenteric plexus, the submucosal plexuses show broad species differences. They are involved in the control of reabsorption and fluids secretion as well as blood flow in the mucosal layer. Moreover, they create connections with other types of ENS plexuses and supply cells to the immune and entero-endocrine systems [17,18].

The enteric neurons are a structure characterized by a high degree of functional autonomy. They control both motor processes and processes related to the secretion and resorption of both gastric contents and the digestive enzymes [18]. A characteristic feature of enteric neurons is their neuronal plasticity, which can be defined as an adaptive variability in the synthesis of biologically active substances and their release into the surrounding environment in response to pathological factors [19].

Hyperglycemia is a condition of elevated blood glucose levels and occurs most often in the course of undiagnosed or poorly treated diabetes. A consequence of prolonged hyperglycemia is damage to many organs and tissues [18]. One of the tissues that are most vulnerable is nerve tissue, and such damage may affect both somatic and autonomic nerves. Dysfunction of the nervous system supplying the gastrointestinal tract is referred to as gastroenteropathy, and it can affect any section of the gastrointestinal system. The consequences of enteric neuronal damage are manifold and are manifested by a wide spectrum of clinical symptoms, which include nausea, the feeling of severe constipation, diarrhea, the abnormal passage of intestinal contents, or frequently occurring pain episodes [20,21,22,23]. The pathomechanism of these disorders is diverse. In addition to well-known mechanisms, including the development of abnormal metabolic pathways (sorbitol, polyol pathway), this may involve a change in the amount of biologically active substances synthesized and released by the neural cells of the enteric neurons [23].

The aim of the current study is to determine how hyperglycemia lasting for six weeks affects the quantitative population of 5-HT-synthesising neurons in the porcine small intestinal system. This also determines whether serotonin is a substance involved in the response of the intramural neurons of the gastrointestinal tract to hyperglycemia. The study was conducted using the pig as a model animal due to the numerous similarities between the porcine and human gastrointestinal tract anatomy and physiology. The mechanisms of insulin secretion and the blood supply to the pancreas are also similar in both species. It should also be noted that the pig is also used as a research model in other studies on metabolic disorders [24,25,26].

## 2. Results

### 2.1. Hyperglycemia

All pigs that were used in the present investigation developed diabetes within approximately 7 days. The main criterion for the development of diabetes was significant hyperglycemia. The mean glucose level before the injection of streptozotocin in both groups remained at the physiological level (5 mmol/L) (Figure 2A). From the first week of the experiment, a significant increase in glucose concentration was observed in all pigs in the experimental group (Figure 2B). The highest glucose concentration was observed in the second week of the experiment. Then, in the animals of the experimental group, the average concentration of glucose in the blood remained above 20 mmol/L (Figure 2C–G). Such a high concentration was maintained until the end of the experiment. It should be underlined that although blood glucose level in STZ-treated animals was notably higher than in controls, all pigs with hyperglycemia survived the duration of the experiment in good condition, and none of the pigs required exogenous insulin supplementation.

### 2.2. Immunofluorescence

Double-labeling immunohistochemistry revealed that 5-HT was present in the investigated parts of the gut. Their presence was detected in both submucosal plexuses as well as in the myenteric plexus. The number of neurons that express serotonin differed between particular segments of the intestine as well as between individual plexuses.

#### 2.2.1. Duodenum

The population of neurons synthesizing 5-HT in the control animals located in the myenteric plexus was established at the level of 8.34 % (±1.67) (Figure 3 and Figure 4A). The 6-week hyperglycemia led to statistically significant changes in the number of 5-HT-immunoreactive neurons (5-HT-IR) in the myenteric plexus. We have noted an increase in the total number of 5-HT neurons to the level of 14.78 % (±1.90) (Figure 3 and Figure 4B). Within the inner and outer submucosal plexus in the control group population of 5-HT-positive neurons constituted 2.98 % (±0.23) and 4.43 % (±0.56), respectively (Figure 3). Treatment with streptozotocin enhanced the population of neurons expressing 5-HT in both submucosal plexuses. In the case of the inner submucosal plexus, an increase has reached the level of 7.56 % (±1.56) (Figure 3 and Figure 4C,D), while in the outer submucosal plexus, the increase in 5-HT positive neurons was slightly higher (9.48 % (±2.06)) (Figure 3 and Figure 4E,F).

#### 2.2.2. Jejunum

In these parts of the small intestine, 5-HT positive neurons were located in all investigated plexuses. In control animals, the most numerous population was observed in the myenteric plexus, 10.33% (±1.98) (Figure 5 and Figure 6A). In the submucosal plexuses, the population of 5-HT-IR neurons was less numerous. In the inner submucosal plexus, 4.66% (±1.33) of Hu C/D-positive neurons expressed 5-HT simultaneously, while the outer plexus contained only 1.95 % (±0.44) of neurons immunopositive to 5-HT (Figure 5 and Figure 6C,D). Diabetes increased the number of 5-HT positive neurons in the myenteric and inner submucosal plexuses, while in the outer submucosal plexus, changes in the number of 5-HT-IR neurons were not observed (Figure 5 and Figure 6B,D,F). In relation to the myenteric plexus, the population of 5-HT positive neurons increased to 15.69 % (±2.34), while in the inner submucosal plexus, the level of 5-HT immunopositive neurons was estimated at 7.89 % (±1.87).

#### 2.2.3. Ileum

In the control group, neurons containing serotonin were present in the myenteric plexus at 3.87% (±0.56) and inner submucosal plexus at 2.04% (±0.23) (Figure 7 and Figure 8A,C), while in the outer submucosal plexus, 5-HT-positive neurons were not observed (Figure 7 and Figure 8E,F). High blood glucose level led to an increase in the number of 5-HT-IR neurons in the myenteric plexus to 5.77% (±0.41) as well as de novo synthesis of 5-HT in neurons forming the outer submucosal plexuses (2.29% (±0.77)) and lack of changes in serotonin expression in the inner submucosal plexus (Figure 7 and Figure 8B–D,F).

#### 2.2.4. Nerve Fibers

Nerve fibers immunoreactive for serotonin were found in all investigated parts of the small intestine. They were present both in the muscle layer as well as in the submucosal area (Figure 9 and Figure 10). In the duodenum muscle layer, the nerve fibers were located between the longitudinal and circular muscle layers. They created long bunds (++), and their density did not change under hyperglycemia conditions (Figure 9A,B). In turn, in the jejunum only single (+) nerve fibers have been observed. Very similar density (+) of serotonin-positive nerve fibers were observed in the ileum muscle layer. In both intestinal areas, differences between control and experimental group were not observed (Figure 9C–F). Microscopic analysis of nerve fibers in the duodenum submucosal layer showed the most numerous population of nerve fibers in the entire area of the small intestines (+++). Their density was the same in the control and experimental group (Figure 10A,B). A slightly less dens network was observed in the jejunum submucosal layer (++) (Figure 10C,D). In turn, the least numerous nerve fibers were noted in the ileum, where we observed only single fibers in both groups (Figure 10E,F). Moreover, in the course of nerve fibers, numerous varicosities were present.

## 3. Discussion

The role of 5-HT in the gastrointestinal tract has been a subject of intense research over many years [15,27]. While its physiological function in secretomotor processes is well understood, its significance in the course of gastrointestinal tract pathological disorders is still not completely known. This study conducted tests aimed at understanding the quantitative changes in the 5-HT positive neurons that are part of the enteric neurons of porcine small intestines in the course of experimentally induced diabetes. The nosological entity that provided the basis for the tests conducted in the study is currently recognized as a condition with an exceptionally dynamic growth trend. As the incidence of diabetes increases, the number of complications often accompanying the disease increases as well [22]. The gastrointestinal tract, due to its complex innervation, is a structure whose proper functioning is impaired in the course of diabetes [21,22,23,28]. These changes are usually caused by disturbances in the normal gastrointestinal tract motility, which converts into the abnormal passage of the gastric content and impairs the resorption and secretion processes within the gastrointestinal tract [29,30,31]. The results obtained in the current study indicate that 5-HT-positive neurons that are part of the small intestinal enteric neurons may be involved in pathological processes of the gastrointestinal tract in the course of diabetes. During the study, a statistically significant increase was observed in the population of 5-HT-synthesising neurons in the myenteric plexuses within all sections of the small intestine. The neurons of this plexus primarily control the intestinal contractile activity, which is clearly impaired in the course of diabetes. The role of 5-HT in this process is multidirectional. It stimulates both the intramural excitatory and inhibitory neurons and is able to induce an effect both increasing and decreasing the intestinal motor activity [5,6,11]. The first effect is achieved by the activation of cholinergic neurons, which increases the release of acetylcholine and causes the contraction of intestinal smooth muscles [32]. The inhibitory effect results from the stimulation of nitrergic neurons, which increases the release of nitric oxide, which, in turn, causes smooth muscle relaxation [32]. Moreover, when the gastric contents reach the intestines, the pressure inside the intestinal lumen increases. This results in an increased release of 5-HT, which then stimulates efferent fibers of the vagus nerve and initiates the intestinal peristaltic reflex [33]. An increased number of 5-HT-positive neurons in the enteric neurons in the course of diabetes may translate into an increase in its concentration in both the intestinal wall and the intestinal lumen, which leads to the intensification of the processes described above. These changes consequently impair motility, which is clinically manifested as alternately occurring diarrhea and constipation episodes that are among the main symptoms of diabetic gastroenteropathy [25]. What is also important is that 5-HT is responsible for the development of the migrating motor complex (MMC) and for normal postprandial intestinal contractions that are often impaired in the course of diabetes [34,35,36]. One of the major disorders in the course of diabetes is the impairment of sensory and secretory functions within the gastrointestinal tract. Serotonin is one of the signaling molecules involved in the transmission of sensory information in the intestinal mucous membrane and a powerful stimulus for increased intestinal juice release [37,38]. The current study demonstrated an increase in the number of neurons in the submucosal ganglia of the small intestine, particularly in the jejunum, and the emergence of a de novo population of 5-HT-positive neurons in the outer submucosal plexus of the ileum. The submucosal plexuses, which are primarily involved in the resorptive and secretory processes through an increase in the 5-HT-positive neuron population, are also involved in changes in adapting the enteric neurons to hyperglycemia. The 5-HT synthesized and released by these changes may modify the intestinal resorption processes. The mechanism of electrolyte secretion regulation by 5-HT is a result of its interaction with the 5-HT2 receptor [39]. The above data indicate that one of the intestinal neuronal responses to hyperglycemia is an increase in the amount of 5-HT, reflected in changes in the number of neurons synthesizing it. Undoubtedly, this results in a significant effect of this amine on shaping both the motor as well as the secretory and sensory processes in the course of diabetes and the often-accompanying gastroenteropathy.

One of the possible reasons for changes in the expression of 5-HT-positive neurons may be an inflammatory process that often develops in the course of diabetes [40]. This process is of particular importance in the context of the development of complications affecting the nervous system, including the autonomic nervous system, which includes the enteric neurons [21,25,40,41]. The main mechanism responsible for the development of inflammation is an increase in the concentration of protein glycation end products, the activation of the receptor for glycation products, the development of oxidative stress, and an increase in the oxygen radical concentration, which results in the activation of the nuclear factor NF-kappaβ and an increase in pro-inflammatory cytokine synthesis [41,42]. The research conducted so far has demonstrated that 5-HT is a molecule whose concentration increases in the course of inflammatory conditions that affect the gastrointestinal tract [15]. The involvement of 5-HT was confirmed in the course of inflammatory processes within the gastrointestinal mucous membrane in the course of inflammatory bowel disease, ulcerative colitis, and Crohn’s disease [14,15,29]. Experimental inflammatory intestinal processes induced by different doses of bisphenol A resulted in an increase in the amount of serotonin in the gastrointestinal tract wall [43,44]. To date, relatively few studies have been dedicated to changes in the expression of 5-HT in the gastrointestinal tract in the course of diabetes. A study conducted on rats with streptozotocin-induced diabetes lasting for 3 and 8 weeks, respectively, showed an increase in the amount of 5-HT in the duodenum and the ileal myenteric plexus [21]. The results presented above confirm the results obtained in the current study.

In conclusion, the current study demonstrated that hyperglycemia lasting for six weeks significantly affects the number of 5-HT-positive neurons within the porcine small intestine. The double immunohistochemical staining method applied in the study enabled the precise determination of quantitative changes within the individual ganglia of the enteric nervous system. In addition, the results of the current study confirm the previous observations concerning the involvement of 5-HT in inflammatory processes, where it acts as an immunomodulatory substance. Moreover, an increase in the number of serotonin-positive neurons may result in an increase in the 5-HT concentration in the gastrointestinal tract wall and affect the motor and secretory processes, which are particularly intense in the small intestines. Further research using the agonists and antagonists of individual 5-HT receptor types may contribute to a better understanding of the precise mechanisms of 5-HT action in the gastrointestinal tract, particularly in the course of diabetic gastroenteropathy.

## 4. Materials and Methods

In total, 10 juvenile pigs of the White Large Polish breed were used in this study. The experiments had been approved by the Local Ethical Committee in Olsztyn (Poland) (decision number 13/2015/DTN) and according to the Act for the Protection of Animals for Scientific or Educational Purposes of 15 January 2015 (Official Gazette 2015, No. 266), corresponding in the Republic of Poland with special attention paid to minimizing any pain and stress reaction. After one week acclimatization period, pigs were randomly divided into two groups: control and experimental with chemically induced diabetes (5 animals in each). Hyperglycemia was induced by a single intravenous injection of streptozotocin under premedication induced by atropine (0.05 mg/kg body weight /BW, given intramuscularly; Atropinum sulf. WZF, Warszawskie Zakłady Farmaceutyczne Polfa S.A., Poland), azaperone (2 mg/kg BW, given intramuscularly Stresnil, Janssen Pharmaceutica, Beerse, Belgium), and streptozotocin (STZ, (Sigma-Aldrich, St. Louis, MO, USA, 0130; 150 mg/kg). Directly before injections, STZ was dissolved in disodium citrate buffer solution (pH = 4.23, 1 g streptozotocin/10 mL solution). The needle was inserted into the ear venous, and the STZ infusion time was about 5 min. In order to eliminate nausea and vomiting after streptozotocin infusion, pigs were fasted for 18 h before the experiment, and the control animals were injected with equal amounts of vehicle (citrate buffer). After diabetes induction, animals were kept in cages tailored to pigs. Animals from both groups receive a standard swine diet (rapeseed meal, 6.0%; soybean meal, 9.0%; wheat, 54.0%; barley, 28.5%; others, 2.5%), and water ad libitum. Blood glucose level was measured before STZ injection, 48 h after induction of diabetes. Next measurements were made in each week of the experiment. Blood glucose concentrations were measured in plasma using an Accent-200 (Cormay, Warsaw, Poland) biochemical analyzer, with the colorimetric measurement at a wavelength of 510 nm/670 nm.

Six weeks after diabetes induction, animals in both groups were deeply anesthetized via intravenous administration of pentobarbital 60 mg/kg body weight (Vetbutal, Biowet, Poland). Afterward, pigs were perfused, and the gastrointestinal tract was prepared as previously described [28]. Next, 2 cm long fragments of the small intestine from the place where nerves from inferior mesenteric ganglia supply the intestine were collected. The samples were postfixed by immersion by the 4% paraformaldehyde for 1 h, rinsed several times with phosphate buffer (PB), and finally transferred to 30% sucrose solution and stored at 4 °C until sectioning. The tissue blocks were cut in frontal or sagittal planes using a Microm HM 560 cryostat (Carl Zeiss, Germany) at a thickness of 12 μm and mounted on gelatinized glass. Subsequently, sections were double-stained by first incubating with primary antisera overnight. Following antibodies were used marker Hu C/D proteins (dilution 1:1000; host rabbit; Invitrogen, Waltham, MA, USA; code A-21271) and serotonin (dilution 1:1000; host rabbit; Zymed Laboratories, San Francisco, CA, USA, code 30778610) were used. After overnight incubation with bovine serum albumin (BSA), the sections were incubated with secondary antibodies Alexa Fluor 488 (dilution 1:1000; host donkey; Invitrogen, USA; code A21202) and Alexa Fluor 546 (dilution 1:1000; host donkey; Invitrogen, USA; code A10040). Both in primary and secondary antibodies, slides were incubated at room temperature. The slides were viewed and photographed using an Olympus BX51 microscope equipped with epifluorescence and appropriate filter sets, coupled with a digital monochromatic camera (Olympus XM 10) connected to a PC and analyzed with Cell Dimension software (Olympus, Tokyo, Japan). Standard controls, i.e., pre-absorption for the serotonin antisera 20 μg per 1 mL of antibody at working dilution. Additionally, omission and replacement of the respective primary antiserum with the corresponding non-immune serum completely abolished immunofluorescence and eliminated specific staining (Figure 11).

### Counting of the Nerve Structures and Statistical Evaluation

To evaluate the percentage of examined neurons, at least 700 Hu C/D-labeled cell bodies in a particular plexus (MP, OSP, and ISP) of each studied animal were examined. Only neurons with well-visible nuclei were counted. To prevent double counting of Hu C/D immunoreactive neurons, the sections were located at least 100 μm apart. The data pooled from all animal groups were statistically analyzed using Statistica 13 software (StatSoft Inc., Tulsa, OK, USA) and expressed as a mean ± standard error (SEM) of mean. Significant differences were evaluated using Student’s t-test for independent samples (* *p* < 0.05, ** *p* < 0.01, and *** *p* < 0.001). Moreover, for semiquantitative evaluation of the density of nerve fibers immunoreactive to each substance studied, an arbitrary scale was used, where (−)—absence of nerve fibers; (+)—single nerve fibers; (++)—rare nerve fibers; (+++) dense nerve fibers.

## Figures and Tables

**Figure 1 ijms-23-04564-f001:**
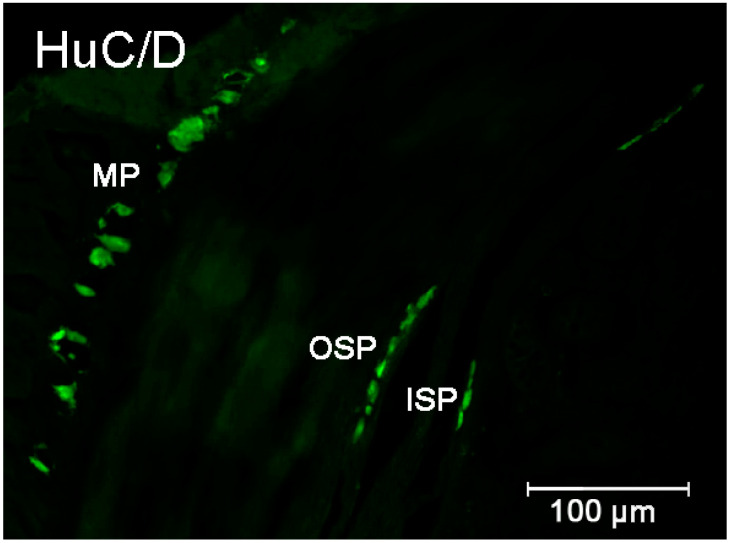
Organization of the enteric neurons in the pig intestine wall demonstrated by the labeling with Hu C/D—used as a pan neuronal marker. Elements of the enteric nervous system: MP—myenteric plexus; OSP—intestinal outer submucosal plexus; ISP—intestinal inner submucosal plexus.

**Figure 2 ijms-23-04564-f002:**
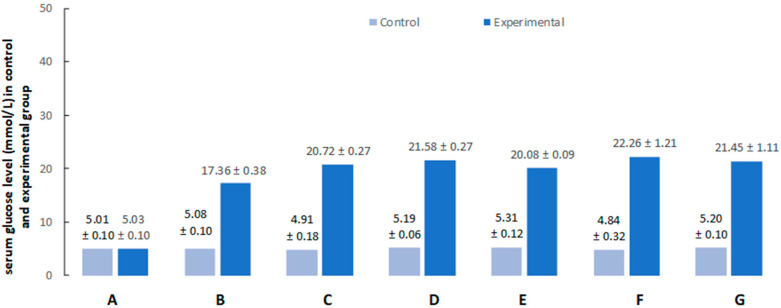
Serum glucose levels after citrate buffer (light blue bars) or streptozotocin (dark blue bars) administration (**A**) at the beginning of the experiment, (**B**) after 1 week, (**C**) after 2 weeks, (**D**) after 3 weeks, (**E**) after 4 weeks, (**F**) after 5 weeks, and (**G**) after 6 weeks.

**Figure 3 ijms-23-04564-f003:**
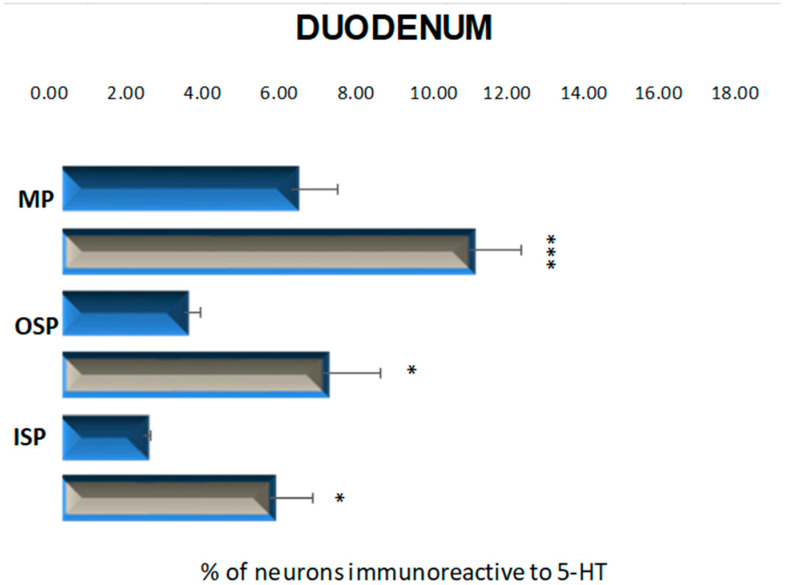
Schematic diagram of the proportion of perikarya immunoreactive to 5-HT of the control (blue bars) and experimental group (grey bars) in the particular parts of duodenum. OSP—the outer submucosal plexus; ISP—the inner submucosal plexus; MP—the myenteric plexus. * *p* < 0.05 and *** *p* < 0.001 indicate differences between all groups for the same neuronal population.

**Figure 4 ijms-23-04564-f004:**
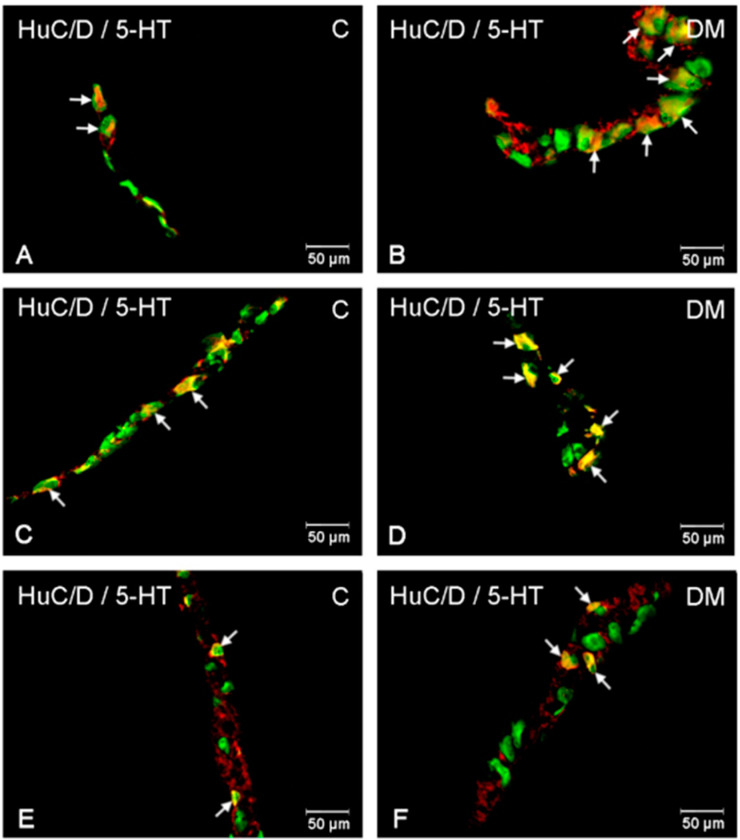
Immunofluorescent microphotographs showing serotonin immunoreactive perikarya in duodenum in the myenteric plexus of the control (**A**) and in experimental group (**B**); inner submucosal plexus of the control (**C**) and in experimental group (**D**); outer submucosal plexus of the control (**E**) and in the experimental group (**F**). (C)—control group; (DM)—diabetes mellitus. Photographs in the right column were created by digital superimposition of two-color channels; Hu C/D (green)- and serotonin-positive (red). The arrows indicated studied cells’ bodies.

**Figure 5 ijms-23-04564-f005:**
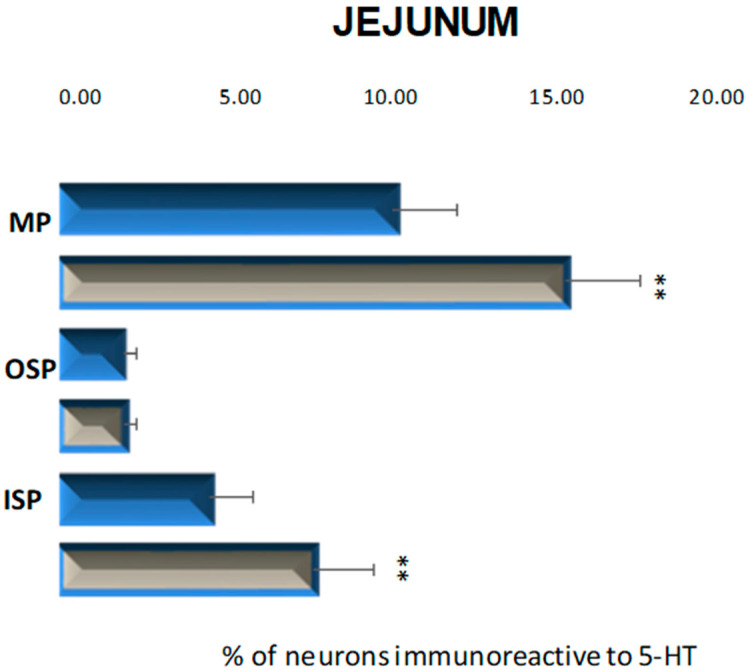
Schematic diagram of the proportion of perikarya immunoreactive to 5-HT of the control (blue bars) and experimental group (grey bars) in the particular parts of jejunum. MP—the myenteric plexus; OSP—the outer submucosal plexus; ISP—the inner submucosal plexus. ** *p* < 0.01 indicates differences between all groups for the same neuronal population.

**Figure 6 ijms-23-04564-f006:**
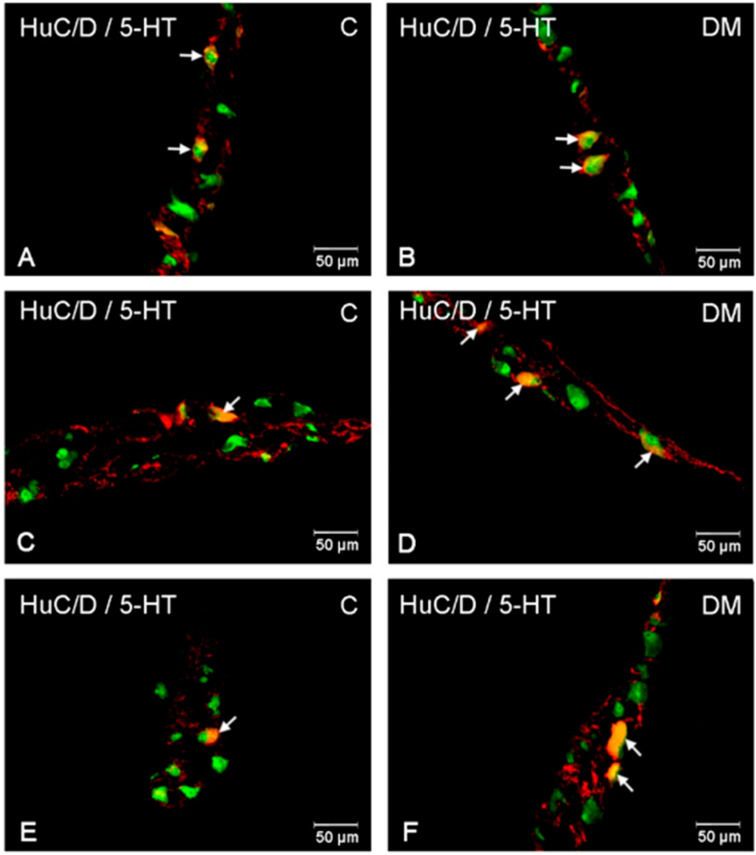
Immunofluorescent microphotographs showing serotonin immunoreactive perikarya in jejunum in the myenteric plexus of the control (**A**) and in experimental group (**B**); inner submucosal plexus of the control (**C**) and in experimental group (**D**); outer submucosal plexus of the control (**E**) and in the experimental group (**F**). (C)—control group; (DM)—diabetes mellitus. Photographs in the right column were created by digital superimposition of two-color channels; Hu C/D (green)- and serotonin-positive (red). The arrows indicated studied cells’ bodies.

**Figure 7 ijms-23-04564-f007:**
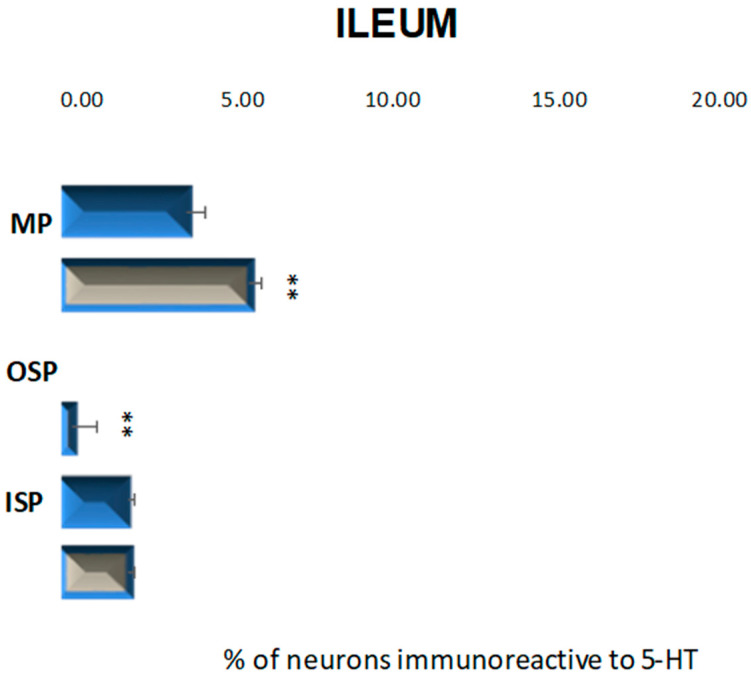
Schematic diagram of the proportion of perikarya immunoreactive to 5-HT of the control (blue bar) and experimental group (grey bars) in the particular parts of ileum. OSP—the outer submucosal plexus; ISP—the inner submucosal plexus; MP—the myenteric plexus. ** *p* < 0.01 indicates differences between all groups for the same neuronal population.

**Figure 8 ijms-23-04564-f008:**
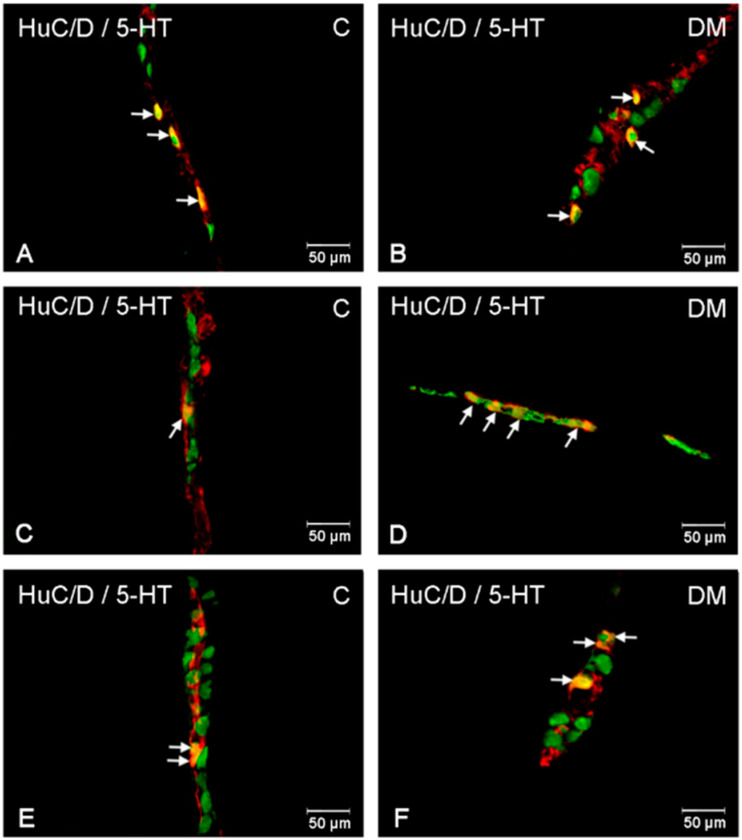
Immunofluorescent microphotographs showing serotonin immunoreactive perikarya in ileum in the myenteric plexus of the control (**A**) and in experimental group (**B**); inner submucosal plexus of the control (**C**) and in experimental group (**D**); outer submucosal plexus of the control (**E**) and in the experimental group (**F**). (C)—control group; (DM)—diabetes m mellitus. Photographs in the right column were created by digital superimposition of two-color channels; Hu C/D- (green) and serotonin-positive (red). The arrows indicated studied cells’ bodies.

**Figure 9 ijms-23-04564-f009:**
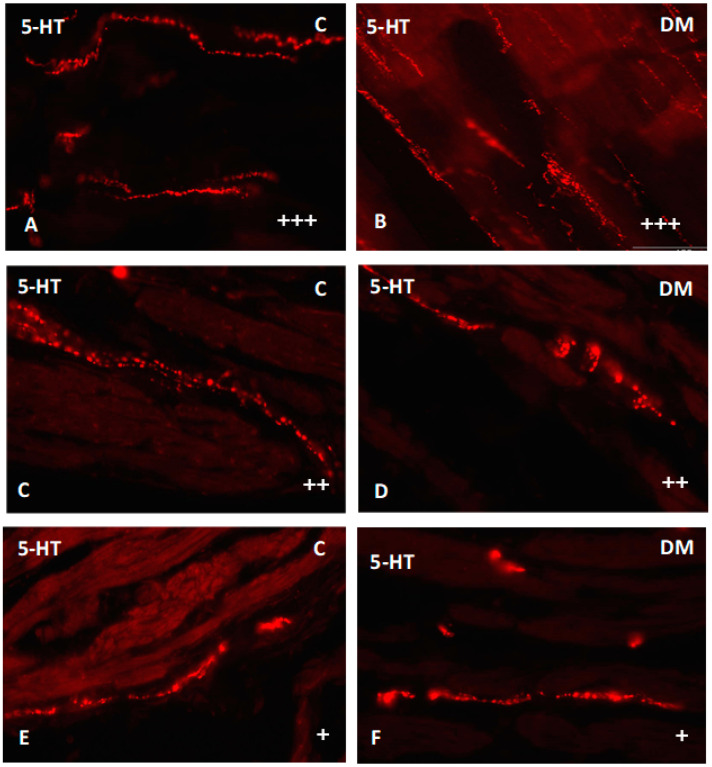
Serotonin-immunoreactive nerve fibers in various parts of mucosal layer in the porcine small intestine. (**A**) The duodenum in control animals (+++); (**B**) the duodenum in experimental group (+++); (**C**) the jejunum in control animals (++); (**D**) the jejunum in experimental group (++); (**E**) the ileum in control animals (+); (**F**) the ileum in experimental group (+). (+)—single nerve fibers; (++)—rare nerve fibers; (+++)—very dense nerve fibers.

**Figure 10 ijms-23-04564-f010:**
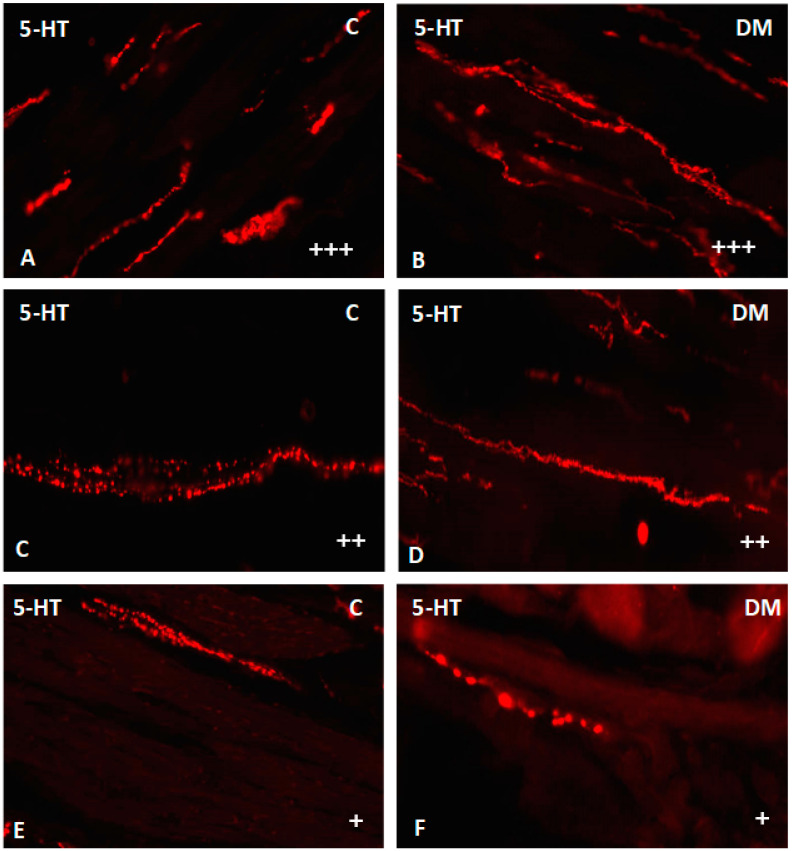
Serotonin-immunoreactive nerve fibers in various parts of submucosal layer in the porcine small intestine. (**A**) The duodenum in control animals (+++); (**B**) the duodenum in experimental group (+++); (**C**) the jejunum in control animals (++); (**D**) the jejunum in experimental group (++); (**E**) the ileum in control animals (+); (**F**) the ileum in experimental group (+). (+)—single nerve fibers; (++)—rare nerve fibers; (+++)—very dense nerve fibers.

**Figure 11 ijms-23-04564-f011:**
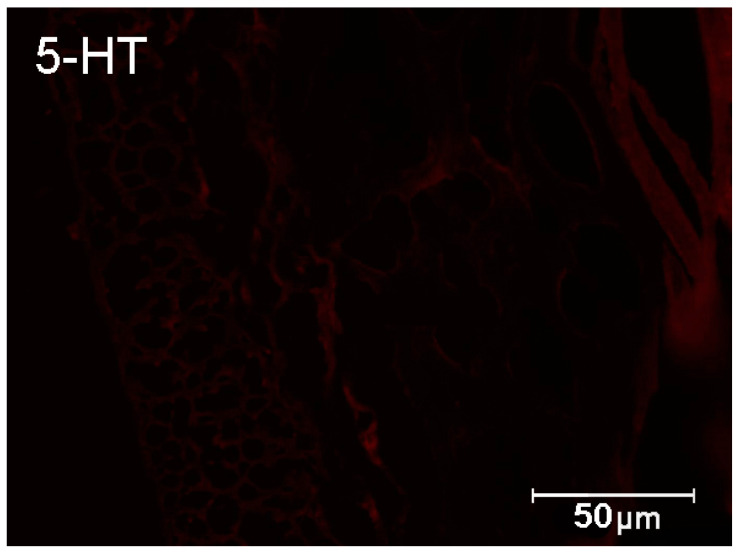
Negative control for 5-HT.

## Data Availability

MDPI Research Data Policies at https://www.mdpi.com/ethics (accessed on 6 March 2022).

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
