# Peer review of "Streptozotocin-Induced Diabetes Causes Changes in Serotonin-Positive Neurons in the Small Intestine in Pig Model"

_ijms, 2022, doi:10.3390/ijms23094564_

Round 1
Reviewer 1 Report
The scientific contents of this manuscript accord with IJMS scopes. The topic of the manuscript is interesting considering the increasing number of diabetic patients with widespread gastrointestinal complications. Because of the different function and structure of the GI segments, a detailed investigation of intestinal region-specific and layer-dependent changes is important. The results are clear, but editing of English language and style is required and major revision is needed.
Major and minor comments:
1. The Abstract section should be reorganized. At the beginning of it, authors should focus to the importance of serotonin in the GI tract instead of general information and highlight its possible susceptibility in diabetic gastroenteropathy. As well as, at the end of this section a conclusion is also missing after the results.
2. In the Introduction part they wrote that in mammals, the majority of 5HT is found in the intestine. It is right, but they should better emphasized (with reference) that the intestinal 5HT production is originated mainly from the enteroendocrine cells, and only small amount of 5HT produced by enteric neurons. Despite of the serotonergic subpopulation is a small part of the enteric neurons, changes in their number has a great impact.
3. Types of plexuses are distinguished in the Introduction, but their function and different roles in enteric regulation are lacking from here. Considering the goal of the study, this information should be included in that part and not only in the discussion.
4. The abbreviation IBD should be given in Introduction part. Otherwise, Crohn’s disease is also an inflammatory bowel disease, so this sentence should be reconsidered.
5. The expression ‘enteric intestinal system’ is meaningless, both enteric and intestinal words refer to gut. (Introduction)
6. I recommend to use the following expressions (instead of others):
immune cells (immune system cells)
enteric neurons (enteric nervous system cells)
7. There are also some meaningless or confused parts in the Results section:
(2.1) ‘STZ treatment animals’ are STZ-treated animals; ‘all pigs with experimental group’ should be all pigs of experimental group or all pigs with hyperglycemia; ‘chronic hyperglycemia…was higher than in controls’ should be blood glucose level….was higher than in controls. Here (but also in several places), the punctuation marks are missing at the end of the sentences.
(Fig. 1) The SEM values are missing from the columns. If the ‘F’ means the glucose level after 5 weeks, where is the final glucose values after 6 weeks of the experiment (before killing)?
(Fig. 2, Fig. 4, Fig. 6) The meaning of the axles should be given in the diagram.
As well as, if authors want to show the percentage of 5HT-IR neurons per total neuronal number, do not write number of 5HT-IR neurons in the legend; maybe using of ‘proportion’ should be practical. And only the relevant significant differences should be written in the figure legends.
(Fig. 3, Fig. 5, Fig. 7) ‘C and DM’ abbreviations on the figures should be also indicated in the legends.
(2.2) The 2.3, 2.4 and 2.5 parts detail the immunofluorescent results in the different gut regions. Therefore, it would be logical to change their subheadings to 2.2.1 etc.
(2.3) The abbreviation IR should be given before using. Sometimes IR, but another time immunoreactive expression is used. It is the same situation with serotonin/5-HT/5HT and Hu C/D or HuC/D and submucous/submucosal or nervous fibres/nerve fibres.
Using the names of investigated groups are also not consequent: experimental group, diabetes group, STZ treatment animals refer to the same animals. It should be logical to use it consequently. For me, the control group is also one of the experimental groups, and if someone check only a figure, it does not turn out that the experimental group is a diabetic group.
(2.5) ‘Injection of STZ led to increased number of 5HT neurons’…Using ‘chronic hyperglycemia led to…’ should be more suitable.
8. In the Discussion, authors refer only one study published 15 years ago (Chandrashekaran et al 2007), in which increasing proportion of 5HT neurons was observed. They should review the recent literature and give more references with others’ findings. (Overall, only three references were cited from the last 5 years.)
9. Methods and references:
The form of different parameters of reagents and applied antibodies should be standardized.
The fixative’s name is missing.
The abbreviations (PB, BSA) are missing.
The incubations’ temperature is missing.
The year of a reference is missing (ref 5).
Author Response
You will find included corrected version of our manuscript entitled ,, Streptozotocin Inducted Diabetes Causes Changes in Serotonin -positive neurons in the Small Intestine in Pig Model”.
Michał Bulc, Katarzyna Palus, Jarosław Całka, Joanna Kosacka, Marcin Nowicki
We appreciate the thorough review. All text improvements of our manuscript have been done in red font.
Here are correction:
Reviewer 1
Major and minor comments:
- The Abstract section should be reorganized. At the beginning of it, authors should focus to the importance of serotonin in the GI tract instead of general information and highlight its possible susceptibility in diabetic gastroenteropathy. As well as, at the end of this section a conclusion is also missing after the results.
Authors answer:
Thank you for your comments. As suggested by the reviewer the abstract was reorganized.
- In the Introduction part they wrote that in mammals, the majority of 5HT is found in the intestine. It is right, but they should better emphasized (with reference) that the intestinal 5HT production is originated mainly from the enteroendocrine cells, and only small amount of 5HT produced by enteric neurons. Despite of the serotonergic subpopulation is a small part of the enteric neurons, changes in their number has a great impact.
Authors answer:
Thank you for your comments. As suggested by the reviewer we added more information concerning enteroendocrine cells as well as appropriate literature positions.
- Types of plexuses are distinguished in the Introduction, but their function and different roles in enteric regulation are lacking from here. Considering the goal of the study, this information should be included in that part and not only in the discussion.
Author answer:
Thank you for your comments. Information including more detailed description of particular plexuses was added.
- The abbreviation IBD should be given in Introduction part. Otherwise, Crohn’s disease is also an inflammatory bowel disease, so this sentence should be reconsidered.
Author answer:
Thank you for your comments. The abbreviation was added and sentence was changed.
- The expression ‘enteric intestinal system’ is meaningless, both enteric and intestinal words refer to gut. (Introduction)
Author answer:
Thank you for your comments. The name has been corrected and standardized.
- I recommend to use the following expressions (instead of others):
immune cells (immune system cells)
enteric neurons (enteric nervous system cells)
Author answer:
Thank you for your comments. Nomenclature has been corrected as suggested by the reviewer
- There are also some meaningless or confused parts in the Results section:
(2.1) ‘STZ treatment animals’ are STZ-treated animals; ‘all pigs with experimental group’ should be all pigs of experimental group or all pigs with hyperglycemia; ‘chronic hyperglycemia…was higher than in controls’ should be blood glucose level….was higher than in controls. Here (but also in several places), the punctuation marks are missing at the end of the sentences.
Author answer:
Thank you for your comments. This paragraph has been improved.
(Fig. 1) The SEM values are missing from the columns. If the ‘F’ means the glucose level after 5 weeks, where is the final glucose values after 6 weeks of the experiment (before killing)?
Author answer:
Thank you for your comments. Figure showing blood glucose level has been corrected and SEM value was added.
(Fig. 2, Fig. 4, Fig. 6) The meaning of the axles should be given in the diagram.
As well as, if authors want to show the percentage of 5HT-IR neurons per total neuronal number, do not write number of 5HT-IR neurons in the legend; maybe using of ‘proportion’ should be practical. And only the relevant significant differences should be written in the figure legends.
Author answer:
Thank you for your comments. Figures have been corrected.
(Fig. 3, Fig. 5, Fig. 7) ‘C and DM’ abbreviations on the figures should be also indicated in the legends.
Author answer:
Thank you for your comments. These abbreviations have been explained.
(2.2) The 2.3, 2.4 and 2.5 parts detail the immunofluorescent results in the different gut regions. Therefore, it would be logical to change their subheadings to 2.2.1 etc.
Author answer:
Thank you for your comments. It has been corrected according to reviewer suggestion.
(2.3) The abbreviation IR should be given before using. Sometimes IR, but another time immunoreactive expression is used. It is the same situation with serotonin/5-HT/5HT and Hu C/D or HuC/D and submucous/submucosal or nervous fibres/nerve fibres.
Author answer:
Thank you for your comments. It has been corrected according to reviewer suggestion.
Using the names of investigated groups are also not consequent: experimental group, diabetes group, STZ treatment animals refer to the same animals. It should be logical to use it consequently. For me, the control group is also one of the experimental groups, and if someone check only a figure, it does not turn out that the experimental group is a diabetic group.
Author answer:
Thank you for your comments. We decided to standardized names of investigated groups and use term “experimental group”.
(2.5) ‘Injection of STZ led to increased number of 5HT neurons’…Using ‘chronic hyperglycemia led to…’ should be more suitable.
Author answer:
Thank you for your comments. It has been corrected according to reviewer suggestion.
- In the Discussion, authors refer only one study published 15 years ago (Chandrashekaran et al 2007), in which increasing proportion of 5HT neurons was observed. They should review the recent literature and give more references with others’ findings. (Overall, only three references were cited from the last 5 years.)
Author answer:
Thank you for your comments. We added literature on the subject from recent years to the manuscript. All new literature position are indicated by red numbers.
- Methods and references:
The form of different parameters of reagents and applied antibodies should be standardized.
The fixative’s name is missing.
The abbreviations (PB, BSA) are missing.
The incubations’ temperature is missing.
The year of a reference is missing (ref 5).
Author answer:
Thank you for your comments. It has been corrected.

Reviewer 2 Report
Thank you for the chance to read your manuscript.
Can you provide example images of the myenteric plexus, the inner and outer submucosal plexus? This would help to visualise the data summarised in your schematic diagram in Fig 2.
In Figure 1, can you show the separate data points for the glucose levels? For C-F, the SEM is either very small (which would be demonstrated in the individual data points), or is missing. Can this be addressed, please.
Can you suggest a further method of quantification of serotonin that is not reliant upon immunofluorescence? Can you also please provide viual examples of the controls mentioned in the materials and methods (standard controls etc for the immunofluorescence)
Author Response
You will find included corrected version of our manuscript entitled ,, Streptozotocin Inducted Diabetes Causes Changes in Serotonin -positive neurons in the Small Intestine in Pig Model”.
Michał Bulc, Katarzyna Palus, Jarosław Całka, Joanna Kosacka, Marcin Nowicki
We appreciate the thorough review. All text improvements of our manuscript have been done in red font.
Here are correction:
Reviewer 2
Can you provide example images of the myenteric plexus, the inner and outer submucosal plexus? This would help to visualise the data summarised in your schematic diagram in Fig 2.
Author answer:
Thank you for your comments. Picture showing enteric neurons organization has been added as figure 1.
In Figure 1, can you show the separate data points for the glucose levels? For C-F, the SEM is either very small (which would be demonstrated in the individual data points), or is missing. Can this be addressed, please.
Author answer:
It has been corrected according to reviewer suggestion.
Can you suggest a further method of quantification of serotonin that is not reliant upon immunofluorescence? Can you also please provide viual examples of the controls mentioned in the materials and methods (standard controls etc for the immunofluorescence)
Author answer:
Thank you for your comments.
You can use semi-quantitative methods such as Western Blot or quantitative methods for determining the amount of mRNA (RT-PCR). But due to the abundant expression of serotonin in the intestinal endocrine system, these methods will not give us an accurate answer to the question of whether the quantitative changes relate to changes in neurons or other cells that produce serotonin in the intestinal wall.
Standard controls, i.e. pre-absorption for the serotonin antisera (20 μg of appropriate antigen per 1 mL of corresponding antibody at working dilution, as well as omission and replacement of the respective primary antiserum with the corresponding non-immune serum in the case of our study completely abolished immunofluorescence and eliminated specific staining.
Photograph from negative control has been added to manuscript.

Round 2
Reviewer 1 Report
Authors’ revision significantly improved the quality of the manuscript; however, some suggestions have forgotten to be corrected or made some new mistakes in this version. Therefore, Authors should check the manuscript more thoroughly and prepare the corrections.
Major comment:
(Fig 3) Authors write in the text that in the duodenum an increase of 5HT-IR neurons was observed (e.g., ~8% in control MP and ~15% in diabetic MP), however figure 3 and its legend demonstrate this result like a decrease, if control means the blue bar and diabetic means the grey bar. Similar problem is visible in the case of OSP and ISP. And only the relevant significant differences should be written in the figure legends (e.g., ** significance value is not in Figure 3, but authors wrote about it in the legend, see also Fig 5 and 7).
Minors:
Using of serotonin or 5-HT expressions in the Abstract should be also consistent.
Correction of ‘enteric nervous system cells’ to ’enteric neurons’ was not happened.
Similarly, ’STZ-treated animals’ instead of ’STZ treatment animals’ or ’blood glucose level….was higher than in controls’ instead of ‘chronic hyperglycemia…was higher than in controls’ …These were not corrected yet. I recommend checking again this paragraph in the Results section.
(Fig 6 and 8) There is a spelling mistake: diabetes mellitus instead of mellites.
(4.1) Using of nerve fibre and nervous fibre is still not consistent.
Dilatation is not equal with dilution, check again the materials and methods section thoroughly.
Author Response
You will find included corrected version of our manuscript entitled ,, Streptozotocin Inducted Diabetes Causes Changes in Serotonin -positive neurons in the Small Intestine in Pig Model”.
Michał Bulc, Katarzyna Palus, Jarosław Całka, Joanna Kosacka, Marcin Nowicki
We appreciate the thorough review. All text improvements of our manuscript have been done in blue font.
Here are correction:
Reviewer 1
Major comment:
(Fig 3) Authors write in the text that in the duodenum an increase of 5HT-IR neurons was observed (e.g., ~8% in control MP and ~15% in diabetic MP), however figure 3 and its legend demonstrate this result like a decrease, if control means the blue bar and diabetic means the grey bar. Similar problem is visible in the case of OSP and ISP. And only the relevant significant differences should be written in the figure legends (e.g., ** significance value is not in Figure 3, but authors wrote about it in the legend, see also Fig 5 and 7).
Author answer:
Thank you for your comments. It has been corrected.
Minors:
Using of serotonin or 5-HT expressions in the Abstract should be also consistent.
Author answer:
Thank you for your comments. It has been corrected.
Correction of ‘enteric nervous system cells’ to ’enteric neurons’ was not happened.
Author answer:
Thank you for your comments. It has been corrected. In all text we used term enteric neurons instead enteric nervous system
Similarly, ’STZ-treated animals’ instead of ’STZ treatment animals’ or ’blood glucose level….was higher than in controls’ instead of ‘chronic hyperglycemia…was higher than in controls’ …These were not corrected yet. I recommend checking again this paragraph in the Results section.
Author answer:
Thank you for your comments. It has been corrected.
(Fig 6 and 8) There is a spelling mistake: diabetes mellitus instead of mellites.
Author answer:
Thank you for your comments. It has been corrected.
(4.1) Using of nerve fibre and nervous fibre is still not consistent.
Author answer:
Thank you for your comments. It has been corrected.
Dilatation is not equal with dilution, check again the materials and methods section thoroughly.
Author answer:
Thank you for your comments. It has been corrected.

Round 3
Reviewer 1 Report
Authors’ revision significantly improved the quality of the manuscript; however, some suggestions have forgotten to be corrected or made some new mistakes in this version. Therefore, Authors should check the manuscript more thoroughly and prepare the corrections.
Major comment:
(Fig 3) Authors write in the text that in the duodenum an increase of 5HT-IR neurons was observed (e.g., ~8% in control MP and ~15% in diabetic MP), however figure 3 and its legend demonstrate this result like a decrease, if control means the blue bar and diabetic means the grey bar. Similar problem is visible in the case of OSP and ISP. And only the relevant significant differences should be written in the figure legends (e.g., ** significance value is not in Figure 3, but authors wrote about it in the legend, see also Fig 5 and 7).
Minors:
Using of serotonin or 5-HT expressions in the Abstract should be also consistent.
Correction of ‘enteric nervous system cells’ to ’enteric neurons’ was not happened.
Similarly, ’STZ-treated animals’ instead of ’STZ treatment animals’ or ’blood glucose level….was higher than in controls’ instead of ‘chronic hyperglycemia…was higher than in controls’ …These were not corrected yet. I recommend checking again this paragraph in the Results section.
(Fig 6 and 8) There is a spelling mistake: diabetes mellitus instead of mellites.
(4.1) Using of nerve fibre and nervous fibre is still not consistent.
Dilatation is not equal with dilution, check again the materials and methods section thoroughly.
Author Response

(The authors gave the same response as above.)
